# Uncertainty-aware Carbon Accounting for Large-scale AI models with Market-based Attribution

## Abstract

The computational demands of large-scale AI models raise significant concerns about their carbon footprint. Current carbon accounting methods for large-scale AI models suffer from three key limitations: they overlook embodied carbon (from hardware manufacturing) or model it simplistically, rely on location-based carbon attribution that fails to reflect individual corporate efforts to decarbonize (e.g., via Power Purchase Agreements (PPAs)), and are deterministic, ignoring inherent uncertainties. This paper proposes CarbonPPA, an uncertainty-aware carbon accounting model with market-based attribution for large-scale AI models. CarbonPPA integrates market-based carbon intensity to accurately account for the impact of PPAs and employs probabilistic modeling to capture uncertainties in the carbon accounting for AI models arising from spatiotemporal variations in manufacturing and operation, as well as evolving efficiency. We develop a comprehensive carbon dataset by aggregating related data from diverse sources, and then we implement a simple yet effective Kernel Density Estimate (KDE) on the distribution of the parameters from the collected dataset. We compare CarbonPPA with LLMCarbon, the state-of-the-art carbon accounting model. The deviation of the accounting result is significant, reaching up to 251.58%

## 1 Introduction

Large-scale AI models have shown remarkable effectiveness in diverse applications (e.g., Natural Language Processing (Shaikh et al., 2024), Video (Comanici et al., 2025), Robot (Etukuru et al., 2025), etc.). However, the increasing scale of model parameters and training data sharply raises computational demand, resulting in considerable carbon footprints. In alignment with the United Nations' Sustainable Development Goals, the AI community is increasingly focusing on decarbonization of society (Wu et al., 2022; Fortier et al., 2025).

Carbon accounting, which quantifies a product's carbon footprint, is essential for assessing environmental impacts and guiding carbon reduction strategies. This quantification process enables organizations to establish emission reduction targets, ensure regulatory compliance, and showcase their dedication to sustainability. For large-scale AI models, the total footprint includes two components: operational carbon arising from electricity use during the operation of AI models, and embodied carbon associated with the manufacture of AI hardware that runs the AI models.

Although recent studies have begun to quantify the carbon footprint of large-scale AI models (Wu et al., 2022; Faiz et al., 2024), current methods have three key limitations. First, they emphasize operational emissions while omitting embodied emissions or modeling them with oversimplified assumptions. In particular, many approaches apply coarse, class-level averages from Life Cycle Assessment (LCA) reports to represent embodied carbon. As power grids decarbonize and data centers procure carbon-free energy, operational carbon is likely to fall, implying that embodied emissions will account for an increasing share of the total footprint of large-scale AI models. Second, current methods all rely on location-based carbon attribution,

which assigns a uniform, grid-average carbon intensity to all electricity consumers within the grid. They fundamentally fail to account for individual corporate efforts to decarbonize, such as investments in renewable energy via Power Purchase Agreements (PPAs). These approaches overestimate the carbon footprint of the AI model run by the entities that invest in renewables via PPAs. Conversely, they underestimate the carbon of the AI models that run by the entities without PPAs. This underestimation occurs because the location-based method fails to account for the residual grid mix, which is the energy sources mix remaining after PPA-contracted renewable energy is subtracted. As renewable energy procurement through PPAs increases, the residual grid becomes more carbon-intensive, leading to a growing disparity between the location-based average and the actual carbon intensity applicable to non-purchasers. Third, existing carbon models for AI are deterministic and thus fail to represent inherent uncertainty in the carbon footprint of large-scale AI models. Key sources of uncertainty include: (1) geotemporal manufacturing variability: a hardware product instance can be fabricated in diverse time periods and from different regions, resulting in different embodied carbon, due to the spatial-temporal dynamics in the carbon intensity of electricity consumed in the manufacturing process; (2) dynamic manufacturing evolution: annual PPAs volume changes, yield and energy efficiency improvement across time will affect the embodied carbon; (3) dynamic operating context: the operational carbon of AI models can vary significantly depending on when and where the operation occurs as the carbon intensity of electricity varies in the spatial-temporal dimension.

This paper proposes CarbonPPA, an uncertainty-aware carbon accounting model with market-based attribution for large-scale AI models to capture the effect of PPAs on carbon accounting and generate probabilistic accounting outcomes. Specifically, we introduce the market-based carbon intensity to integrate PPAs into the carbon footprint accounting of large-scale AI models. Then, we develop parameter models for individual hardware components, such as processors, memory, and storage, that contribute to the AI models' embodied carbon. To estimate the parameter distributions, a comprehensive hardware and electricity dataset is constructed by integrating information from multiple sources. These include Environmental, Social, and Governance (ESG) reports from hardware suppliers, statistical data from grid operators, industry analyses, and peer-reviewed literature, as summarized in Table 1. A straightforward yet efficient distribution modeling approach is employed: collected data are first converted into frequency histograms and subsequently processed using Kernel Density Estimation (KDE) to derive continuous probability density functions for the parameters.

We evaluate the performance of CarbonPPA against LLMCarbon, the state-of-the-art carbon accounting approach for large-scale AI models. The evaluation is performed based on four representative large AI models (XLM, T5, GPT-3, and Switch) (Wu et al., 2022; Patterson et al., 2021). We compare the performance at key distribution percentiles (minimum, 20th, 50th, 80th, and maximum) of CarbonPPA's probabilistic outputs against LLMcarbon. The results show that the deviations are significant, reaching up to 251.58% for the embodied carbon and 138.76% for the operational carbon, which highlight the critical importance of uncertainty-aware modeling for the carbon footprint of large-scale AI models.

Our contributions are summarized as follows:

- We propose a new uncertainty-aware carbon accounting model with market-based attribution for large AI models. This model produces distribution-based estimates of carbon footprint rather than point estimates, enabling AI company to incorporate risk assessment into sustainability decisions.

- We formalize and integrate market-based carbon attribution into AI carbon accounting, moving beyond the conventional location-based method. This allows for a more accurate reflection of an AI company's individual renewable energy investments (via PPAs).

- We make an effort to develop a comprehensive carbon dataset comprising PPAs data, and AI hardware-related parameters across multiple technology nodes, drawing on diverse sources including technology reports, ESG and LCA reports, as well as electricity carbon intensity data from 256 regional grid operators. To support the decarbonization of the AI industry, we will open-source the dataset and codes (see Supplementary Material).

## 2 BACKGROUND ON CARBON ATTRIBUTION

To lower carbon emissions, firms begin to invest in renewables via PPAs (e.g., Intel purchased 80% of its annual electricity from renewables in 2021 (statista., 2025)), which confer renewable energy credits and reduce reported emissions. Under Scope 2 guidance (Agency, 2021), carbon attribution follows two approaches. The location-based method assigns all consumers the same average grid mix, crediting renewables to the grid overall and not to individual investors. The market-based method lets consumers claim the environmental attributes of contracted renewables (e.g., PPAs). Uncovered demand is assigned the residual mix, which excludes electricity already claimed through contracts. Consequently, market-based attribution yields consumer-specific carbon intensities, while location-based attribution yields a uniform intensity within a regional grid. For more details, please refer to the Appendix A.1.

## 3 METHODOLOGY

### 3.1 THE BOUNDARY OF CARBONPPA

We examine the carbon footprint across four standard lifecycle stages: (1) Hardware manufacturing; (2) Hardware transport; (3) Operational use, covering emissions resulting from software execution, primarily due to electricity consumption; and (4) End-of-life processing. Among these, the operational use phase corresponds to the operational carbon of AI models, while the remaining stages contribute to their embodied carbon. Manufacturing and operational use are the dominant sources of emissions, accounting for the vast majority (Alissa et al., 2025). While others make negligible contributions and are therefore omitted. Within manufacturing, we focus on the main emissions from materials, energy, and chemical gases, while excluding ancillary components such as buildings, cooling infrastructure, and human labor.

### 3.2 MARKET-BASED CARBON INTENSITY

When certain renewable energy sources are contracted out via PPAs, the electricity source mix that remains within the grid is referred to as the residual grid mix. The carbon intensity associated with this residual mix is the residual carbon intensity ($CI_{res}$) (Maji et al., 2024). Consider a grid where $ef^k$ denotes the constant carbon emission factor of source $k$ (e.g., wind), $E^k$ represents the electricity generated by source $k$, of which $E_{ppa}^k$ represents the portion contracted under PPAs. Let $E$ denote the total electricity generation in the grid, and $E_{ppa}$ be the total PPA-contracted electricity; it follows that $\sum E_{ppa}^k = E_{ppa}$. Under the market-based attribution, entities holding PPAs can claim the low-carbon benefits associated with their procured electricity. As a result, all renewable energy covered by PPAs is subtracted prior to computing $CI_{res}$. Following that, we have Eq. 1. Then, the carbon intensity of a consumer using the market-based attribution ($CI_m$) in that grid can be calculated by Eq. 2

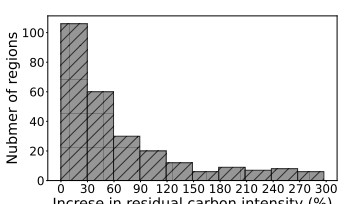

Figure 1: Average increase in residual carbon intensity across 265 regions worldwide in 2024 when all renewable energy is contracted out.

$$CI_{res} = \frac{\sum_{k \in \mathcal{E}} ef^k \cdot (E^k - E_{ppa}^k)}{E - E_{ppa}} \quad (1) \qquad CI_m = CI_{res} \cdot (1 - f_{ppa}) \quad (2)$$

where $f_{ppa} \in [0, 1]$ denotes the proportion of a consumer's electricity supply covered by PPAs. Unlike the location-based attribution, the market-based carbon intensity varies across consumers within the same grid. Consumers without PPAs (i.e., $f_{ppa} = 0$) have a carbon intensity $CI_m = CI_{res}$, whereas those able to meet their entire electricity demand via PPAs achieve $CI_m = 0$. This approach systematically benefits investors

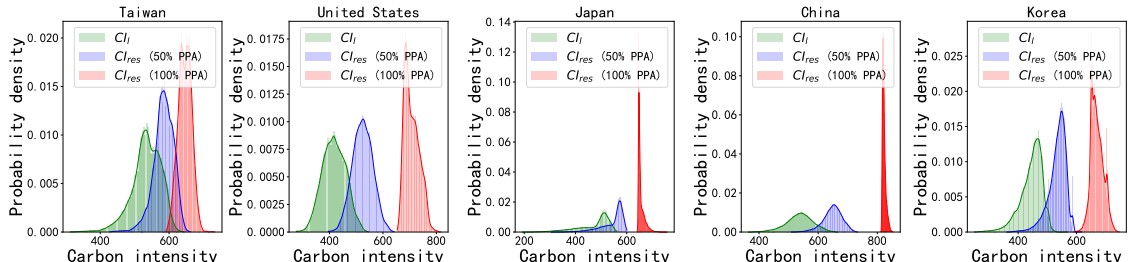

Figure 3: Histograms of hourly location-based carbon intensity and residual carbon intensity from 2021 to 2024 in five major IC production regions with their individual kernel density estimates.

in renewable energy over non-investors. For more about location-based carbon intensity, please refer to Appendix A.2.

Figure 1 presents a histogram showing the relative increase in $CI_{res}$ compared to $CI_l$ across 265 global regions in 2024, under the scenario where all renewable energy sources are contracted out (data source: ElectricityMaps (Maps., 2025)). Regions with greater integration of solar and wind power exhibit more significant rises in $CI_{res}$. As renewable energy continues to expand in power systems worldwide, coupled with the growing procurement of electricity through PPAs, this gap is anticipated to widen further in the future. Within a given region, $CI_{res}$ rises in proportion to the volume of renewable energy procured through purchases. Changes in carbon intensity over a typical week are illustrated for Germany as renewable procurement increases (see Figure 2). The grid derives a significant proportion of its electricity from solar generation, resulting in considerably lower grid carbon intensity during daylight hours. However, as the share of PPAs grows, not only does $CI_{res}$ increase, but the temporal fluctuation in carbon

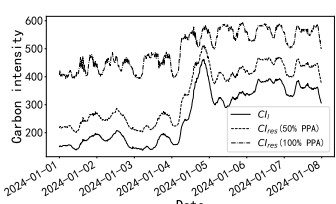

Figure 2: Weekly trace showing how residual CI in Germany differs from location-based CI as more and more renewables are contracted out.

intensity also becomes less pronounced. Figure 3 shows histograms of hourly location-based carbon intensity and residual carbon intensity data (with 50% and 100% PPAs contracted out) from 2021 to 2024 in five major IC production regions with their individual kernel density estimates, where we can also find that the difference between location-based and market-based carbon intensity is significant.

### 3.3 Embodied carbon modeling

We model the embodied carbon of AI models ($EC_{model}$) from three key components as Eq.3 shows, i.e., the carbon caused by processors ($EC_{model}^p$), memory ($EC_{model}^m$), and storage ($EC_{model}^s$).

$$EC_{model} = EC_{model}^p + EC_{model}^m + EC_{model}^s \tag{3}$$

### 3.4 Embodied carbon of AI models associated with processors

The embodied carbon of AI models associated with processors used are modeled based on the following components: (a) the operational duration of AI models on the processors, represented as $t_p$, relative to the processor's total lifetime $T_p$; (b) carbon resulting from electricity consumption during hardware fabrication, determined by multiplying the electricity consumed Per unit of die Size ($EPS$) by the market-based carbon intensity of the manufacturing electricity supply, $CI_m$; (c) carbon from raw material usage, quantified as

$MPS$ (Material required Per Size); (d) emissions from specialty gases such as fluorinated compounds, expressed as $GPS$ (Gas emitted Per Size); (e) the processor's die size, denoted as $S$; and (f) the fabrication yield $Y$. Then, the embodied carbon of processors attributable to AI models can be formulated as follows:

$$EC^p_{model} = \frac{t_p \cdot S}{T_p \cdot \tilde{Y}} \cdot (\tilde{CI_m} \cdot \tilde{EPS} + GPS + MPS) \tag{4}$$

Here, we denote $\tilde{CI_m}$ as the probabilistic model of market-based carbon intensity $CI_m$. Similarly, we have $\tilde{Y}$ and $\tilde{EPS}$. A detailed characterization of uncertainty for these parameters will be provided below.

*Market-based carbon intensity distribution.* In embodied carbon accounting, $CI_m$ represents the carbon intensity of electricity used in semiconductor manufacturing. This key factor depends on the composition of energy sources (e.g., solar, wind, etc.) for power generation, along with related PPAs. The carbon intensity varies spatiotemporally, influenced by production timelines, geographic placement of manufacturing sites, and different PPAs. A major source of uncertainty stems from temporal fluctuations across yearly cycles, which arise from seasonal changes in renewable energy supply and variations in electrical demand and PPAs. Figure 4 shows the carbon intensity of processors manufactured by TSMC in Taiwan and Intel in the USA with different carbon attribution, which is derived from PPAs data and carbon intensity data from 2021 to 2024.

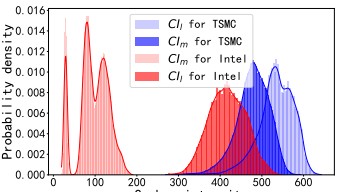

Figure 4: Location and market-based $CI$ for processors.

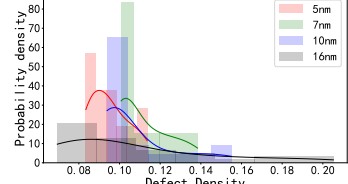

Figure 5: Defect Density (D) for processors.

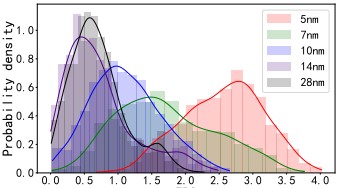

Figure 6: Energy per size (EPS) for processors.

*Yield Distribution.* Semiconductor yield is the fraction of good, defect-free dies on a wafer relative to the total die count. This parameter is inherently uncertain, driven mainly by time-varying defect densities (D) across fabs. To study this systematically, we analyze TSMC's historical defect density records for four technology nodes (Cutress, 2020). We construct defect density histograms and apply Kernel Density Estimation (KDE) (Vanderplas, 2024) to obtain probability density functions of the defect density distributions, as shown in Figure 5. Then we can compute yields using the Poisson yield mode: $Y = e^{(-S \cdot D)}$ (EESemi, 2005).

*EPS Distribution.* Uncertainty in $EPS$ primarily arises from temporal fluctuations in energy efficiency across semiconductor fabrication stages. We construct $EPS$ distributions using $EPS$ estimates from the STEC model (Zhang et al., 2024), ACT model (Gupta et al., 2022), and imec model (Boakes et al., 2023), together with annual efficiency improvement data reported in TSMC's ESG disclosures (TSMC, 2023a) for multiple technology nodes (e.g., 5 nm, 10 nm). The workflow comprises three transformations: (a) normalize per-node energy efficiency over time relative to a baseline year; (b) adjust raw $EPS$ values by dividing by these normalized efficiency to reflect technological progress; and (c) model the distribution by implementing a dual-stage distribution modeling approach, i.e., converting the processed data into frequency histograms, then applying KDE to obtain continuous probability density functions of $\tilde{EPS}$, shown in Figure 6.

### 3.5 EMBODIED CARBON OF AI MODELS ASSOCIATED WITH MEMORY

We model the embodied carbon of AI models associated with memory from following components: (a) the fraction of the device's lifetime utilized by the model, captured by the ratio of memory runtime $t_m$ to

memory lifetime $T_m$; (b) memory fabrication-related emissions attributable to electricity use, computed by multiplying the electricity per unit size ($EPS$) by the market-based carbon intensity during manufacturing ($CI_m$) and then divided by bit density ($BD$); (c) emissions released independent of electricity, including materials, packaging, denoted $\alpha_m$; and (d) the installed memory capacity $C_m$. Then, the embodied carbon of AI models associated with memory can be modeled as follows:

$$EC_{model}^m = t_m/T_m \cdot C_m \cdot (\tilde{CI_m} \cdot EPS/BD + \alpha_m) \tag{5}$$

Unlike processors, where manufacturing is often concentrated at a single foundry (e.g., most GPUs at TSMC in Taiwan), memory is produced by multiple vendors (e.g., SK Hynix, Samsung, Micron, etc.). When the fabrication location is uncertain, we model the region as a discrete random variable. Regional probabilities are assigned in proportion to each area's share of global IC capacity for the relevant process node (Bhagavathula et al., 2024), using capacity splits reported by industry sources (BCG, 2024) as weights. We then construct a composite carbon intensity distribution for each node via a mixture approach that integrates: (a)regional distributions: the distributions developed based on historical carbon intensity data for each major manufacturing region and PPAs data of each manufacturer from 2021 to 2024; (b) capacity-weighted sampling: a Monte Carlo sampling strategy where region selection follows normalized capacity shares; (c) mixture aggregation: combining the sampled observations across regions and applying kernel smoothing to obtain the final distribution.

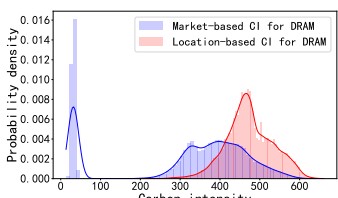

Figure 7: CI for memory.

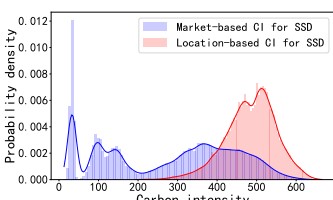

Figure 8: CI for SSD.

This framework captures uncertainty from PPAs, geographical production distributions, and temporal energy mix variations. As a result, Figure 7 compares the market-based and location-based CI for memory. We can find obvious differences between the two attributions.

### 3.6 Embodied carbon of AI models associated with storage

We model the embodied carbon of AI models associated with storage from following components: (a) the utilization ratio of the storage's lifetime, given by $t_s/T_s$, where $t_s$ is the model's storage usage time and $T_s$ is the storage lifetime; (b) manufacturing emissions of storage from to electricity consumed, computed as $EPG \cdot CI_m$, where $EPG$ is the electricity consumed per GB during fabrication and $CI_m$ is the market-based carbon intensity; (c) the emissions released independent of electricity (e.g., materials), denoted as $\alpha_S$ and available from industry reports (SEAGATE, 2024); and (d) installed storage capacity $C_s$. Then, the embodied carbon of AI models associated with storage can be calculated as Eq. 6. Similarly, we can also get the distribution of market-based and location-based CI for storage as Figure 8, where differences between the two attributions are also obvious.

$$EC_{model}^s = t_s/T_s \cdot C_s \cdot (\tilde{CI_m} \cdot EPG + \alpha_S) \tag{6}$$

### 3.7 Operational carbon

The operational carbon of an AI model ($OC_{model}$) is the emissions attributable to the electricity it consumes during operation. It depends on the model's operational electricity consumption ($E_o$) and the market-based carbon intensity of the supplied electricity ($CI_m$), and can be expressed as:

$$OC_{model} = \tilde{CI_m} \cdot E_o = \tilde{CI_m} \cdot \sum_{i \in HardwareSet} (P_i \cdot eff_i \cdot n_i \cdot t_i) \cdot PUE \tag{7}$$

Table 1: The data source of the carbon dataset we developed for CarbonPPA

| Parameter | Description | Unit | Source |
|---|---|---|---|
| CI | Carbon Intensity data across 256 regions | g/kWh | ElectricityMaps(Maps., 2025) |
| $eff^k$ | carbon emission factors | g/kWh | Research paper (Zhang et al., 2024) |
| PPAs | Power Purchase Agreements data | % | Industrial reports (statista., 2025) ESG reports (Micron, 2025) (Kioxia, 2024; Seagate, 2024; Samsung, 2025; TSMC, 2024; hynix, 2025) |
| Fabrication capacity | Global wafer fabricatioin capcity by regions | % | Industrial report (BCG, 2024) |
| Energy effciency | Annual improvement of procees energy effciency | % | ESG reports (TSMC, 2023b; 2022; 2021) |
| EPG | Electricity consumed per GB | kWh/GB | LCA reports (SEAGATE, 2024; Hynix, 2021) |
| Die size | 51 GPUs and 50 CPUs | mm2 | Industrial reports (Techpowerup, 2024) |
| Process nodes | 51 GPUs and 50 CPUs | nm | Industrial reports (Techpowerup, 2024) |
| Defect density | Defect density trend across time | % | Industrial reports(Anandtech, 2020) |
| EPS | Electricity consumed per die Size | kWh/cm2 | Research paper (Gupta et al., 2022; Zhang et al., 2024; Boakes et al., 2023) |
| GPS | Emission from Gas per die Size | g/cm2 | (Gupta et al., 2022; Zhang et al., 2024) |
| MPS | Emission from Material used per die Size | g/cm2 | dustrial reports (Boyd, 2011) |
| BD | Bit density | GB/cm2 | Industrial reports (Choe, 2017) |

where $\tilde{CI}_o$ denotes the probability distribution of the electricity's carbon intensity, reflecting uncertainty in when and where training or inference occurs. $P_i$ is the peak power of hardware $i$; $eff_i$ is its efficiency (obtainable from hardware efficiency models (Faiz et al., 2024)); $n_i$ is the count of units of hardware $i$; and $t_i$ is the runtime on hardware $i$, which can be estimated using FLOP-based models (Faiz et al., 2024).

## 4 EVALUATION

We make an effort to develop a hardware-electricity dataset by aggregating data from diverse sources, including ESG reports disclosures from hardware vendors, statistics released by power system operators, industry reports, and peer-reviewed literature, as summarized in Table 1. We compare CarbonPPA against LLMCarbon, the current state-of-the-art carbon model for large-scale AI, using publicly available training data from four representative models: XLM, T5, GPT-3, and Switch (Wu et al., 2022; Patterson et al., 2021). Evaluation is conducted by comparing CarbonPPA's probabilistic outputs with LLM-Carbon at major distribution percentiles (min., 20th, 50th, 80th, and max.) based on the collected carbon dataset. At present, the PPAs data of hardware manufacturers is available, but detailed data on the residual electricity mix within power grids is often unavailable for each region, reflecting a common real-world limitation. We

Table 2: The comparison between CarbonPPA and LLM-Carbon on embodied carbon accounting.

| Hardware inforamtion | | | |
|---|---|---|---|
| Hardware | Die size/Unit | Number | Technology |
| CPU | 1.47 cm2 | 512 | 12nm |
| GPU | 8.15 cm2 | 64 | 16nm |
| Storage | 32TB | 64 | SSD |
| Memeory | 256GB | 64 | 10nm ddr4 |

| Accounting result | | | | | |
|---|---|---|---|---|---|
| Component | Model | Embodied Carbon (kg) at each Percentile | | | |
| | | Min. | 20th | Median | 80th | Max. |
| Total | CarbonPPA | 138.60 | 179.43 | 238.37 | 295.40 | 498.11 |
| | LLMCarbon | | | 271.01 | | |
| | Deviation$\Delta$ | 53.45% | 26.64% | 68.24% | 108.50% | 251.58% |
| CPU | CarbonPPA | 0.97 | 1.27 | 1.48 | 1.81 | 2.84 |
| | LLMCarbon | | | 1.16 | | |
| | Deviation$\Delta$ | 91.0% | 88.3% | 86.3% | 83.3% | 73.8% |
| GPU | CarbonPPA | 83.67 | 111.15 | 133.50 | 168.88 | 329.2 |
| | LLMCarbon | | | 138.39 | | |
| | Deviation$\Delta$ | 33.4% | 11.5% | 6.3% | 34.4% | 162.0% |
| Memory | CarbonPPA | 5.43 | 5.83 | 8.54 | 10.73 | 18.27 |
| | LLMCarbon | | | 10.53 | | |
| | Deviation$\Delta$ | 95.7% | 95.4% | 93.2% | 91.5% | 85.5% |
| Storage | CarbonPPA | 48.22 | 61.14 | 94.84 | 113.98 | 147.88 |
| | LLMCarbon | | | 120.94 | | |
| | Deviation$\Delta$ | 61.62% | 51.34% | 24.51% | 9.28% | 17.70% |

set the $CI_{res} = CI_l$, which means PPAs in the power grids are relatively small, to show the effect of AI hardware manufacturers' investment in renewable energy on the carbon footprint of AI models in Section 4. In Section 4.2, we assume half of the renewable energy in power grids is contracted out to show the effect of residual carbon intensity on the carbon footprint of AI models. In section 4.3, we show the impact of different PPAs ratios on the carbon footprints of AI models.

### 4.1 EMBODIED CARBON EVALUATION

We evaluate the embodied carbon accounting performance of CarbonPPA and LLMCarbon using the publicly available XLM training dataset (Wu et al., 2022), which, to our knowledge, represents the only publicly available source disclosing hardware-level embodied carbon data for large-scale AI model training. The setup

Table 3: The comparison between CarbonPPA and LLMCarbon on operational carbon accounting

| AI Models | Training information | Accounting Models | Operational Carbon at each Percentile (ton) | | | | | | | | | |
| | | | Spatial Dimension | | | | | Temporal Dimension (ton) | | | | |
| | | | Min. | 20th. | Median | 80th. | Max. | Min. | 20th. | Median | 80th. | Max. |
| XLM | Tarinin Day: 20.4; PUE: 1.1;Ave. Power: 342 kw; Num. of device: 512 | LLMCarbon | | | 36.58 | | | | | 24.29 | | |
| | | CarbonPPA | 2.87 | 23.44 | 45.80 | 59.63 | 86.42 | 9.10 | 25.87 | 34.76 | 39.44 | 52.95 |
| | | Deviation # | 92.16% | 35.91% | 25.21% | 63.00% | 136.24% | 62.54% | 6.49% | 43.11% | 62.38% | 117.99% |
| T5 | Tarinin Day: 20; PUE: 1.12;Ave. Power: 310 kw; Num. of device: 512 | LLMCarbon | | | 33.10 | | | | | 21.98 | | |
| | | CarbonPPA | 2.58 | 21.34 | 41.63 | 54.32 | 77.90 | 8.14 | 23.70 | 30.93 | 35.95 | 48.52 |
| | | Deviation # | 92.22% | 35.54% | 25.76% | 64.10% | 135.35% | 62.96% | 7.81% | 40.71% | 63.56% | 120.77% |
| GPT3 | Tarinin Day: 14.8; PUE:1.1;Ave. Power: 330 kw; Num. of device:10K | LLMCarbon | | | 500.11 | | | | | 332.15 | | |
| | | CarbonPPA | 39.05 | 319.61 | 628.57 | 820.78 | 1194.08 | 125.07 | 353.74 | 466.46 | 542.76 | 734.21 |
| | | Deviation # | 92.19% | 36.09% | 25.69% | 64.12% | 138.76% | 62.34% | 6.50% | 40.44% | 63.41% | 121.05% |
| Switch | Tarinin Day: 27; PUE:1.1;Ave. Power: 245 kw; Num. of device: 1K | LLMCarbon | | | 67.74 | | | | | 44.99 | | |
| | | CarbonPPA | 5.31 | 43.67 | 86.03 | 111.05 | 161.24 | 16.67 | 48.07 | 64.26 | 72.61 | 97.92 |
| | | Deviation # | 92.17% | 35.53% | 27.00% | 63.94% | 138.02% | 62.96% | 6.84% | 42.83% | 61.40% | 117.64% |

involves 64 servers (details in Table 2) used over 20.4 days, with hardware assumed to have a 5-year lifetime (Wu et al., 2022).

As shown in Table 2, a marked contrast exists between the single-point deterministic model (LLMCarbon) and the probabilistic CarbonPPA model. LLMCarbon estimate consistently falls within the distribution provided by CarbonPPA. CarbonPPA reveals a much wider potential range, frequently showing that emissions can be over 250% higher at the maximum percentile compared to the LLMCarbon estimate. Significant deviations are also observed at the component level. For GPUs and storage, the dominant sources of uncertainty, CarbonPPA yields a range of 83.67–329.2 t and 48.22–147.88 t, respectively, while LLMCarbon gives a simple average value of 138.39 t and 120.94 t, corresponding to maximum deviations of 162.0% and 61.62%.

We can find that LLMcarbon gives a relatively large estimate in the distribution of total embodied carbon (271.01t vs. 238.37t at the median). The reason behind this phenomenon is that many hardware manufacturers invest in renewables through PPAs, thereby reducing the carbon intensity of their electricity consumption. CarbonPPA can capture this effect on the operational carbon accounting of large-scale AI models, whereas LLMcarbon lacks this capability. Our evaluation also reveals critical limitations in deterministic accounting and emphasizes the need for probabilistic approaches. We recommend that environmental assessments for AI systems report uncertainty intervals to better inform sustainable development practices.

## 4.2 OPERATION CARBON EVALUATION

We compare CarbonPPA with LLMCarbon on the operational carbon accounting for four large-scale AI models, XLM, T5, GPT3, and Switch, using published training details (Patterson et al., 2021) as shown in Table 3. We evaluate along two dimensions: (i) spatial variation: fixing the training date and varying location across 265 regions (data from ElectricityMpas (Maps., 2025)); and (ii) temporal variation: fixing location (California) while varying time (2021–2024).

CarbonPPA offers a comprehensive distribution of operational carbon across both spatial and temporal dimensions, whereas LLMCarbon produces only an average single-point estimate. As shown in Table 3, the estimates from LLMCarbon exhibit substantial deviations from those of CarbonPPA across all evaluated models and percentiles. These relative deviations are especially pronounced at upper percentiles, frequently surpassing 135% in the spatial dimension and 117% in the temporal dimension. Even at lower percentiles, deviations remain notable, generally approximating 90% spatially and 60% temporally. For example, in the spatial dimension for GPT-3, CarbonPPA yields a range of 39.05t–1194.08t across percentiles, while LLMCarbon gives a single value of 500.11t, leading to a relative deviation of 138.79% maximum percentile. We can find that LLMcarbon always gives a relatively small estimate in the distribution of operational carbon for each large-scale AI mode. The reason behind this phenomenon is that renewable energy in the grid has been contracted out via PPAs, resulting in a higher carbon intensity in the residual electricity. CarbonPPA can capture the effect of PPAs on the operational carbon accounting of large-scale AI models, whereas LLMcarbon lacks this capability.

### 4.3 THE EFFECT OF PPAs

In this section, we examine how PPAs affect carbon accounting for a large-scale AI model. We study the impact of $f_{ppa}$ of an organization that operates an AI model and $CI_{res}$ of the grid supplying electricity separately. Figure 9 presents the operational carbon emissions from GPT-3 training within the California grid under the scenario where excluded PPA-contracted renewable energy is not considered (i.e., $CI_{res} = CI_l$). A substantial discrepancy between location-based and market-based carbon accounting methods is evident. For example, at 90% PPA coverage, the location-based median can be nine times the market-based value (345.43 kg vs. 34.54 kg). The results indicate that operational carbon emissions decrease as the share of PPAs increases, suggesting that investing in renewable energy via PPAs can effectively reduce the carbon footprint of AI models. Figure 9 illustrates the operational carbon emissions when an AI operator does not invest in renewable energy (i.e., $f_{ppa} = 0$), under varying proportions of excluded PPA-contracted renewable energy (i.e., $CI_{res} > CI_l$) in the California grid. It can be observed that the difference between location-based and market-based emissions remains significant. For instance, at 90% excluded PPAs, the median differs by 96.81% (267.91 kg vs. 527.26 kg). Furthermore, operational carbon emissions increase as the fraction of excluded PPA-contracted renewable energy rises.

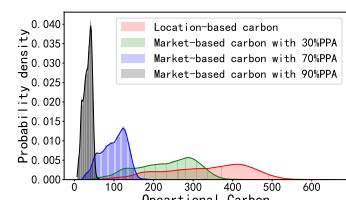

Figure 9: The distribution of operation carbon over PPAs.

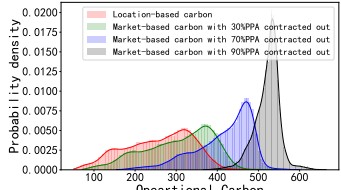

Figure 10: The distribution of operation carbon over PPAs excluded.

We have more discussion about insights and use cases for CarbonPPA in Appendix A.3.

## 5 RELATED WORK

Current carbon accounting methods for AI models primarily focus on operational emissions, calculated as the product of electricity consumption and grid carbon intensity. Most of the existing literature is dedicated to tracking or estimating electricity consumed. Several studies have introduced software-based tools to monitor CPU/GPU power consumption in real-time during model training or inference (Henderson et al., 2020; Anthony et al., 2020; Budennyy et al., 2022; Fu et al., 2025), while others estimate energy use based on hardware specifications such as thermal design power (Lannelongue et al., 2021). Yet these frameworks consistently overlook the embodied carbon emissions originating from AI hardware infrastructure, which represent a significant portion of the total footprint, especially as grids decarbonize and data centers adopt more renewable energy. For embodied carbon, current methods like SustainableAI (Wu et al., 2022) rely on coarse-grained manufacturer-reported average emission factors for hardware components. LLMCarbon (Faiz et al., 2024) employs a deterministic parametric model for processors while using averaged data from ESG reports for memory and storage devices. Yet these approaches are deterministic and location-based, neglecting the role of PPAs and the inherent uncertainty in the carbon footprint of large AI models.

## 6 CONCLUSION

In this paper, we propose CarbonPPA, an uncertainty-aware carbon accounting model with market-based attribution for large-scale AI models. CarbonPPA integrates PPAs into the accounting process and can capture the uncertainty in both operational and embodied carbon, enabling an AI company to make risk-aware decisions to meet sustainable targets.

## REPRODUCIBILITY STATEMENT

We have included the source codes and datasets as supplementary materials to submit.

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

# A  APPENDIX

## A.1  MORE ON BACKGROUND

**Large-scale AI model carbon footprint.** The carbon footprint of large-scale AI models includes two categories: operational carbon and embodied carbon. Operational carbon is from the electricity consumed during model deployment. Deploying a large-scale AI model demands not only considerable electricity but also significant computational hardware, such as high-performance GPUs (e.g., NVIDIA A100 (Nvidia, 2024)) and storage. Embodied carbon is the emissions produced during the manufacturing of the hardware required to run these AI models. The processors supporting large-scale AI models are often fabricated using cutting-edge semiconductor processes (such as 5nm technology), which contribute substantially to the embodied carbon of AI models. As the share of renewable energy in power grids increases and more data centers transition to carbon-neutral power sources, the operational carbon emissions from running AI models are expected to decrease. Consequently, the embodied carbon will represent a growing share of AI models' total carbon footprint (Gupta et al., 2022; Zhang et al., 2024).

**Carbon Attribution.** To reduce carbon footprint, some companies are shifting electricity use to regions or periods with low grid carbon intensity, while many companies are investing in renewable energy via PPAs. PPAs are usually long-term contracts for renewable energy between a consumer and an electricity producer, wherein the consumer can claim renewable energy credits for their investment and lower the emissions caused by the electricity they consume. Under the Scope 2 GHG Protocol guidance (Agency, 2021), carbon attribution to consumers can follow two distinct approaches, depending on how renewable generation and the underlying grid mix are allocated. The location-based carbon attribution assigns an identical electricity mix to all consumers within a defined geographic area. Under this approach, green energy is credited to the grid as a whole, and the carbon intensity is calculated based on the average emissions of the entire grid mix, incorporating both renewable and non-renewable sources in proportion to their actual generation. Importantly, this method does not account for individual green energy investments made by specific consumers. Instead, any renewable energy contributions are shared collectively among all consumers in the region. The market-based carbon attribution enables consumers who invest in renewable energy to claim the environmental attributes of that electricity and account for lower carbon emissions, even if the physical power they consume comes from the grid, which comprises both renewable and non-renewable sources. For any remaining electricity demand not covered by such investments, or for consumers without renewable contracts, carbon emissions are calculated using the residual grid mix. This residual mix excludes all electricity that has been claimed under contractual instruments. The market-based attribution allows electricity to be attributed according to investment sources, resulting in varying carbon intensities across different consumers.

## A.2  LOCATION-BASED CARBON INTENSITY

The location-based carbon intensity ($CI_l$) of electricity is defined as the carbon emission rate (in g/kWh) during power generation. This intensity is calculated as the weighted average of the carbon intensities of all contributing energy sources, based on their respective shares of electricity generation. The mathematical formulation for the carbon intensity is as follows:

$$CI_l = \frac{\sum ef^k \times E^k}{\sum E^k} \tag{8}$$

where $ef^k$ and $E^k$ represent the carbon emission factor and the electricity generated by energy source $k$, respectively.

## A.3 INSIGHTS AND POTENTIAL USE CASE FOR CARBONPPA

The development and evaluation of CarbonPPA reveal several critical insights into carbon accounting for large-scale AI models and open up a range of practical applications for stakeholders across the AI ecosystem.

### A.3.1 KEY INSIGHTS

**The Impact of Market-Based Attribution:** Our results highlight the significant influence of corporate renewable energy investments, particularly through PPAs, on carbon accounting. The location-based method, which ignores such investments, systematically misrepresents the carbon footprint of both green and non-green energy consumers. CarbonPPA's market-based approach not only rewards decarbonization efforts but also exposes the growing carbon intensity of residual grids, offering a more equitable and accurate accounting framework.

**Uncertainty-aware Accounting:** CarbonPPA demonstrates that uncertainty is not merely a statistical nuance but a central feature of carbon accounting in the large-scale AI models. The significant deviations (up to 251.58% in embodied carbon and 138.76% in operational carbon) between probabilistic and deterministic models underscore the risk of relying on point estimates. By providing distributional outputs, CarbonPPA enables AI developers and operators to quantify and communicate the variability and reliability of their carbon footprints, facilitating more robust sustainability planning and risk management.

**Embodied Carbon Cannot Be Ignored**: As operational carbon decreases due to grid decarbonization and efficiency gains, embodied carbon becomes an increasingly dominant component of the total footprint. CarbonPPA's detailed modeling of hardware components, processors, memory, and storage shows that embodied emissions are subject to significant spatial, temporal, and technological uncertainties. Ignoring these factors, as current methods often do, leads to incomplete and potentially misleading environmental assessments.

### A.3.2 POTENTIAL USE CASES

**AI Company Sustainability Strategy:** CarbonPPA can help AI companies make informed decisions regarding hardware procurement, energy sourcing, and operational scheduling. By simulating different PPA scenarios and hardware choices under uncertainty, companies can optimize their carbon budgets, set risk-aware targets, and report sustainability metrics with confidence intervals, enhancing transparency and credibility.

**Policy and Regulation Support:** Regulators and standard-setting bodies can use CarbonPPA as a benchmark for developing more nuanced carbon accounting guidelines for the AI industry. The model's ability to differentiate between market-based and location-based emissions can inform policies that incentivize renewable energy investments and penalize carbon-intensive operations.

**Green AI Model Development:** Researchers and engineers can integrate CarbonPPA into the AI development lifecycle to evaluate the environmental impact of training scheduling and deployment options. By incorporating carbon awareness early in the design process, the community can foster the creation of lower-carbon AI systems.

**Third-Party Auditing and Certification:** CarbonPPA's open-source dataset and probabilistic framework provide a foundation for independent verification and certification of AI carbon footprints. Auditors can use the tool to validate corporate sustainability claims and issue certifications that reflect both average emissions and associated uncertainties.

**Investment and ESG Reporting for AI company:** Investors and stakeholders increasingly demand accurate environmental performance data. CarbonPPA enables AI companies to report carbon footprints in a manner

that reflects their actual renewable energy investment and operational contexts, supporting better ESG ratings and more aligned investment decisions.

### A.4 THE USE OF LARGE LANGUAGE MODELS

We used large language models to polish and improve the fluency of the writing. All ideas, analyses, and conclusions remain our own.

