# OpenReview forum: "Uncertainty-aware Carbon Accounting for Large-scale AI models with Market-based Attribution"
_ICLR.cc/2026/Conference — ICLR 2026 Conference Withdrawn Submission_

### Official Review · Reviewer_xchG · 2025-10-31

**Soundness:** 2
**Presentation:** 3
**Contribution:** 3
**Rating:** 4
**Confidence:** 4

**Summary:**

This paper proposes CarbonPPA, a novel carbon accounting framework for large-scale AI models that incorporates market-based attribution via Power Purchase Agreements (PPAs) and models both embodied and operational emissions probabilistically. The authors compare CarbonPPA against the state-of-the-art deterministic model LLMCarbon and demonstrate significant deviations (up to 251% for embodied carbon and 138% for operational carbon), highlighting the importance of accounting for uncertainty and corporate renewable-energy investments. A comprehensive dataset is compiled from 256 regional grid operators, ESG reports, and industry analyses, and distributional estimates are derived using Kernel Density Estimation (KDE). The work concludes that embodied carbon will dominate future footprints as grids decarbonize, and that uncertainty-aware, market-based accounting is essential for credible sustainability reporting.

**Strengths:**

**Novelty & Relevance** CarbonPPA is the first AI-specific carbon model to combine (i) market-based attribution that rewards PPA investments, and (ii) full probabilistic treatment of both embodied and operational emissions. This directly addresses a pressing gap as AI workloads grow and Scope 2 guidance evolves.

**Rich Empirical Evaluation** The paper evaluates four representative large models (XLM, T5, GPT-3, Switch) across spatial (265 regions) and temporal dimensions, showing large deviations from LLMCarbon. The released dataset (256 regions, 51 GPUs, 50 CPUs, PPA shares, etc.) is a valuable community resource.
Sound Methodology & Reproducibility: KDE-based distributional modeling of key parameters (yield, EPS, CIres) is technically solid; appendices provide full equations, data sources, and open-source code, enabling replication and extensions.

**Weaknesses:**

**Weakness 1. Limited Ablation & Sensitivity Analysis.** While the aggregate deviations are large, the paper does not isolate the marginal contribution of (i) market-based vs. location-based attribution, (ii) embodied vs. operational uncertainty, or (iii) individual distributional assumptions. A factorial or Sobol sensitivity analysis would clarify which modeling choices drive the reported ranges.

**Weakness 2. Scarcity of Real PPA & Residual-Mix Data** For most regions the authors set CIres = CIl due to missing residual-mix data; the “50 % & 90 % PPA” scenarios are illustrative rather than empirical. This weakens the practical claim that CarbonPPA “accurately” reflects corporate renewables investments; results may change once actual residual mixes are published.

**Weakness 3. Narrow Hardware & Manufacturing Coverage** Embodied modeling focuses on processors, DRAM, and SSDs, but omits networking equipment, cooling, buildings, and packaging lines which can be 30–50 % of total embodied emissions (per recent LCA studies). Yield distributions are taken from public defect-density trends rather than fab-specific data, potentially under-estimating true variability.

**Questions:**

see weakness.

---

### Official Review · Reviewer_1a2E · 2025-11-01

**Soundness:** 1
**Presentation:** 2
**Contribution:** 1
**Rating:** 0
**Confidence:** 5

**Summary:**

This paper proposes an uncertainty-aware carbon accounting model with market-based attribution for large-scale AI models. The authors develop a carbon dataset from diverse sources and implement a Kernel Density Estimate (KDE) on the distribution of the parameters from the collected dataset.

**Strengths:**

1)	The decarbonizing problem for large AI models introduced in paper is interesting.
2)	A new comprehensive carbon dataset from diverse sources is introduced.
3)	The proposed KDE-based parameters estimation method seems feasible.

4)    Different features that have impact on the AI model energy consumption are considered in the proposed model.

**Weaknesses:**

1)	Based on what is presented in the paper, it looks more like an industrial report rather than a machine learning research paper.
2)	The method proposed in the paper is rather simple. All the equations used for calculions seem empirical without any theoretical explanation. Therefore, the novelty of this paper is very limited.
3)	The proposed method should be compared against other state-of-the-art baselines in the field, e.g., Bayesian methods and ensembles.
4)	The authors should explain more about the equations presented in Section 3 from a scientific perspective.
5)    The KDE results presented in Figure 5 look poor, some essential parameters are missing, e.g., the bandwidths.

**Questions:**

Overall, I am aware of the potential impact of this work bu I do not think ICLR is the proper venue for this paper.

---

### Official Review · Reviewer_FU8t · 2025-11-03

**Soundness:** 2
**Presentation:** 3
**Contribution:** 1
**Rating:** 2
**Confidence:** 3

**Summary:**

This paper addresses the challenge of embodied carbon accounting for AI models and proposes a more accurate carbon estimation framework. The authors develop carbon accounting models for key hardware components, including processors, memory, and storage devices. The proposed model characterizes the carbon footprint as a probability distribution, explicitly capturing uncertainty in embodied emissions. The framework also incorporates market-based attributions into the accounting model. Furthermore, the authors construct and open-source a hardware–electricity dataset.

**Strengths:**

The paper is well-written.

**Weaknesses:**

- The consideration of embodied carbon accounting is not novel, as similar analyses have been conducted in prior works (e.g., [1, 2]). Although the authors incorporate a market-based attribution approach, the conceptual advancement over existing methods appears incremental.

- The embodied carbon emissions from hardware manufacturing are not unique to AI workloads. It remains unclear whether the authors have quantified what portion of total manufacturing-related emissions can be directly attributed to AI development and deployment. Without this distinction, the relevance of the embodied carbon model specifically to AI systems is limited.

- The hardware–electricity dataset is not derived from real measurement data but instead relies on public sources, particularly the XLM training dataset (Wu et al., 2022). The selection and calibration of parameters are insufficiently justified, raising concerns about whether the parameter choices accurately reflect diverse real-world configurations and usage scenarios.

[1] Mulligan, C. and Elaluf-Calderwood, S., 2022. AI ethics: A framework for measuring embodied carbon in AI systems. AI and Ethics, 2(3), pp.363-375.

[2] Gupta, U., Elgamal, M., Hills, G., Wei, G.Y., Lee, H.H.S., Brooks, D. and Wu, C.J., 2022, June. ACT: designing sustainable computer systems with an architectural carbon modeling tool. In Proceedings of the 49th Annual International Symposium on Computer Architecture (pp. 784-799).

**Questions:**

To comply with Power Purchase Agreements (PPAs), do AI companies need to reduce AI workload demand, limit hardware purchases?

---

### Note · Authors · 2025-11-12

I have read and agree with the venue's withdrawal policy on behalf of myself and my co-authors.